# Liver-Targeted Nanoparticles Facilitate the Bioavailability and Anti-HBV Efficacy of Baicalin In Vitro and In Vivo

**DOI:** 10.3390/biomedicines10040900

**Published:** 2022-04-14

**Authors:** Weiming Xu, Yijun Niu, Xin Ai, Chengjie Xia, Ping Geng, Haiyan Zhu, Wei Zhou, Hai Huang, Xunlong Shi

**Affiliations:** 1Department of Biological Medicines & Shanghai Engineering Research Center of Immunotherapeutics, Fudan University School of Pharmacy, Shanghai 201203, China; 19211030036@fudan.edu.cn (W.X.); 20211030038@fudan.edu.cn (Y.N.); 21211030037@fudan.edu.cn (X.A.); 18211030035@fudan.edu.cn (C.X.); 17211030050@fudan.edu.cn (P.G.); haiyanzhu@fudan.edu.cn (H.Z.); haihuang@fudan.edu.cn (H.H.); 2Department of Chemistry, Fudan University, 220 Han Dan Road, Shanghai 200433, China; zhouw@fudan.edu.cn

**Keywords:** baicalin, liposomes, HBV, bioavailability, liver-targeted drug delivery, apolipoprotein A1

## Abstract

The anti-hepatitis B virus (HBV) efficacy of baicalin (BA) is mediated by HBV-related hepatocyte nuclear factors (HNFs). However, this efficacy is severely limited by the low bioavailability of BA. Therefore, a novel liver-targeted BA liposome was constructed to promote the bioavailability and antiviral ability of BA. The results showed that apolipoprotein A1 (ApoA1)–modified liposomes (BAA1) significantly enhanced BA’s cellular uptake and specific distribution in the liver. Furthermore, the substantial inhibitory effects of BAA1 on HBsAg, HBeAg, HBV RNA, and HBV DNA were assessed in HB-infected cells and mice. Western blotting, co-immunoprecipitation, and transcriptomics analysis further revealed that the enhanced anti-HBV efficacy of BAA1 was attributed to the interaction between hepatocyte nuclear factors (HNFs) and estrogen receptors (ERs). Based on the findings, we propose that the ApoA1-modified liposomes aid BA in inhibiting HBV transcription and replication by augmenting its bioavailability and the HNFs–ERs axis.

## 1. Introduction

Chronic hepatitis B virus (HBV) infection is a common public health problem that is prevalent worldwide and is the leading global cause of liver cirrhosis and hepatocellular carcinoma (HCC) [1]. Approximately 250 million people suffer from chronic HBV infection, and 20% of them have been estimated to die from complications such as liver cirrhosis, liver failure, or HCC [2].

Currently, drugs used clinically to treat HBV infection are mainly divided into two categories: immunomodulators and nucleoside analogs. Immunomodulators, such as interferon and thymosin α-1, have severe adverse reactions and are not suitable for many people. Nucleoside analogs, such as lamivudine and entecavir (ETV), need to be taken for a long period and are prone to viral resistance, relapse after cessation, and poor management of viral proteins [3,4]. Therefore, it is necessary to develop a new therapeutic strategy for HBV.

Baicalin (BA) has been reported to possess many pharmacological activities, such as antitumor, antibacterial, anti-inflammatory, and antiviral activities [5,6]. We have previously proved that BA could inhibit HBV transcription and replication via an HNF4α–HNF1α dependent axis [7]. In addition, our unpublished study found that BA can promote the binding of HNF4α and ERα into a heterodimer by upregulating the phosphorylation of estrogen receptor α (ERα), thereby attenuating the transcriptional activation effect of HNF4α on HNF1α. However, the anti-HBV activity of BA was limited owing to its poor oral bioavailability [8]. The use of liposomes has been suggested as an effective delivery system that improves the bioavailability of BA, and hence its therapeutic value [9]. However, there is little experimental evidence regarding the use of BA liposomes in anti-HBV treatment.

Ligand-modified liposomes can aid in active liver-targeted drug delivery via receptor–ligand interactions [10,11]. Apolipoprotein A1 (ApoA1) is the main protein component of high-density lipoprotein (HDL) and can bind to scavenger receptor type I (SR-B1) on hepatic parenchymal cells [12]. ApoA1-modified liposomes contain anti-HBV iRNA plastids that downregulate the HBV viral protein more significantly than unmodified liposomes [13]. ApoA1-modified cationic liposome containing doxorubicin was developed to treat multidrug-resistant tumors [14]. Therefore, ApoA1 might be used as a specific ligand to achieve liposomal liver-targeted drug delivery.

In this study, a novel ApoA1-modified BA liposome (BAA1) was constructed using effervescent dispersion technology [15] to enhance BA bioavailability. Furthermore, its characteristics and anti-HBV efficacy were explored.

## 2. Materials and Methods

### 2.1. Reagents

Baicalin (purity > 98%) was purchased from ChromaBio, Chengdu, China. Entecavir (purity > 98%) was sourced from MedChemExpress (Jersey, NJ, USA). Hydrogenated soybean lecithin was purchased from AVT (Shanghai, China). Citric acid and Tween 80 were sourced from Shanghai Titan Technology Damas-beta Co., Ltd. Ethanol and other reagents were obtained from Sinopharm Chemical Reagent (Shanghai, China).

### 2.2. Animals

Male BALB/c mice (weight: 18–22 g) were provided by Shanghai SLAC Laboratory Animal Co. Ltd. (Shanghai, China) and kept under stable temperature conditions (26 ± 1 °C) with free access to food and water.

### 2.3. Preparation of Liver-Targeted Baicalin Liposomes (BAA1)

The liposome preparation was performed by using effervescent dispersion technique. Briefly, BA (96 mg), hydrogenated soybean phospholipid (96 mg), Tween 80 (50 mg), and citric acid (50 mg) were used to prepare the BA liposome mixture, as described previously [15]. Finally, 9.6 mg ApoA1 was added to the lipids, and the mixture was continuously stirred until cheese-like consistency was obtained. Then, the sample was collected at −80 °C, freeze-dried in a vacuum freeze-dryer, and stored at −20 °C until further investigation. The ordinary BA liposomes (BALP, without ApoA1) and empty liposomes (Empty LP, without BA) were prepared in the same way.

### 2.4. Physicochemical Characterization of BAA1

The particle size, polydisepersity index (PDI), and zeta potential of liposomes were measured with the Malvern Zetasizer Nano Zs90 Analyzer (Malvern Instruments Ltd., Malvern, UK).

### 2.5. The Entrapment Efficiency of BAA1

Ultrafiltration centrifugation was performed to determine entrapment efficiency. First, BAA1 was dissolved in ultrapure water and placed in an ultrafiltration centrifuge tube (8–10 kDa) at 5000 rpm/min for 30 min to collect the filtrate. The filtrate obtained was dissolved in ultrapure water to which methanol was added, and the mixture was vortexed for 5 min, diluted to the exact multiple, and the BA content was determined by HPLC [16]. All samples were measured on the HPLC system (Shimadzu Fast Liquid Phase System LC-20A) through a reversed-phase C18 column (Symmetry Shield RP18 Columns; 5 μm, 4.6 × 250 mm; Waters, Milford, MA, USA). Acetonitrile and 0.5% (*v*/*v*) aqueous phosphoric acid (35:65, *v*/*v*) were chosen for the mobile phase at the detection wavelength of 278 nm. The flow rate was 1.0 mL/min at 35 °C, and the injection volume was 20 μL.

The following formula were used to calculate the entrapment efficiency: EE% = (W_t_ − W_f_)/W_t_ × 100%. Where EE% is the encapsulation efficiency, W_free_ is the free drug content in the BAA1 sample, and W_total_ is the total drug content in the BAA1 preparation.

### 2.6. In Vitro Release Studies of Liposomes

The dialysis method was employed to evaluate the in vitro release of liposomes. In short, 2 mL of BAA1 (5 mg/mL BA) and BA solution (5 mg/mL BA) were placed in a dialysis bag (molecular weight cutoff 8–10 kDa) and immersed in 200 mL of PBS (pH 7.4) with 100 rpm stirring at 37 °C. At different time points, 1 mL of the dissolution medium was collected and supplemented with the same volume of fresh PBS. Finally, the collected samples were centrifuged at 12,000 rpm for 10 min to detect the BA content by HPLC [9]. For the detection conditions of HPLC, please refer to Section 2.5.

### 2.7. Cell Culture and HBV Transfection

HepG2 cells were seeded in 24-well plates at 1 × 10^5^ cells per well for subsequent viral transfection. HepG2 cells were transfected with the pHBV1.2 plasmid using the Lipo^™^6000 transfection reagent (Beyotime Biotechnology, Shanghai, China) and named pHBV1.2-transfected HepG2 cells. The amount of plasmid added to each well in a 24-well plate was 500 ng. The pHBV1.2 plasmid [7,17] carrying the HBV genome (genotype C) was kindly gifted by Professor Zhenghong Yuan of Fudan University, China. The HepG2 cells were transfected with AAVDJ vectors [18] (PackGene Biotech, Guangzhou, China) carrying the HBV genome (genotype B) and called the AAVDJ-transfected HepG2 cells.

### 2.8. Cell Counting Kit-8 (CCK-8) Assay

Cell viability was determined by Cell counting kit-8 (CCK-8) kit (Beyotime Biotechnology). CCK-8 was added into cell medium and incubated for 2 h to measure cell viability using a microplate reader at OD_450_ nm.

### 2.9. HBsAg and HBeAg Assay In Vitro and In Vivo

The BA and BAA1 solutions were added to the abovementioned transfected HepG2 cells, respectively, and replaced every 2 days. After 96 h of incubation, the cell supernatants were collected for HBsAg and HBeAg analyses by ELISA kit (Shanghai Kehua Bioengineering Co., Ltd., Shanghai, China).

In pHBV1.2-infected mice, HBsAg in the mouse serum was detected by the HBsAg ELISA kit (Shanghai Kehua Bioengineering Co., Ltd.).

### 2.10. Quantitative RT-PCR (qRT-PCR)

The transcriptional levels of HBV-specific transcripts [19] and HNFs in HepG2 cells were evaluated by qRT-PCR and normalized to the housekeeping gene GAPDH. In addition, HBV DNA in the culture supernatant was detected by qPCR [20].Potential plasmid effects were adequately removed by washing with PBS and changing cell culture media. As previously reported [21], total RNA was isolated and reversed to cDNA using TRIzol reagents and the PrimeScript™ RT reagent Kit with gDNA Eraser (TaKaRa, Osaka, Japan). The TB Green^®^ Premix Ex Taq™ (Tli RNaseH Plus) (TaKaRa) and StepOnePlus™ Real-Time PCR System (Thermo Fisher Scientific, Waltham, MA, USA) were used to perform qPCR analysis based on the following thermocycling parameters: 95 °C for 5 min and hold for 40 cycles; 95 °C, 15 s; and 60 °C, 1 min. The PCR primers are given in Appendix A.

### 2.11. Western Blotting

Total cell proteins were obtained with the Western IP lysis buffer (Beyotime Biotechnology). BCA assay, SDS–PAGE, and Western blotting were performed using a standard procedure. The primary antibodies against HNF1α, FOXA2, and ERα were obtained from CST (Boston, MA, USA). The anti-GAPDH antibody was purchased from Beyotime Biotechnology. The anti-HNF4α antibody was obtained from Abcam (Cambridge, UK). The second antibodies HRP-labeled Goat anti-Mouse IgG or HRP-labeled Goat anti-Rabbit IgG were obtained from Beyotime Biotechnology. All antibodies were diluted at the ratio of 1:1000. The protein bands were visualized by ultrasensitive chemiluminescence (ECL; Millipore, MA, USA) with the ChemiDoc XRS^+^ Imager (Bio-Rad, Hercules, CA, USA) and analyzed by Image J software (NIH, Bethesda, MA, USA).

### 2.12. Co-Immunoprecipitation (Co-IP)

After 4 days of BAA1 or BA treatment, the cells were used to obtain the total proteins. Total protein lysate (2 mg) was incubated with 2 μg of anti-HNF4α/anti-ERα antibodies and mixed with 40 μL of protein A/G magnetic beads (Beyotime Biotechnology) overnight at 4 °C. Rabbit IgG served as the negative control. The protein–antibody bead complex was centrifuged at 3000 rpm for 5 min and washed three times with PBS. Finally, possible interactions between estrogen receptor α (ERα) and HNF4α were explored by Western blotting. The protein lysate that was not incubated with specific antibodies was used as the input sample.

### 2.13. Cellular Uptake of BA

pHBV1.2-transfected HepG2 cells were treated with BAA1 or BA for 1, 2, 4, 8, and 12 h, respectively. The cells were repeatedly frozen and thawed three times and mixed with concentrated hydrochloric acid. Then, 50 μL of the mixture was removed to blend with 25 μL of IS (200 ng/mL carbamazepine methanol solution), 100 μL acetonitrile, and 75 μL of saturated ammonium sulfate. The mixture was centrifuged at 12,000 rpm for 10 min, and the supernatant was blow-dried with nitrogen at 36 °C. The residue obtained was reconstituted in methanol for BA detection by LC–MS [22,23,24].

The LC–MS system was made up of a 6500+ QTRAP mass spectrometer (AB SCIEX, USA) equipped with an electrospray ionization (ESI) source system, and an LC-20A HPLC system (Shimadizu, Kyoto, Japan). Components were separated on a reverse-phase C18 column (2.7 μm, 2.1 × 1000 mm, Navigatorsil), eluting with a gradient mobile phase system, which consisted of acetonitrile–water (0–4 min at 25:75, 4–5 min at 25:75 to 40:60, 5–5.1 min at 40:60 to 25:75, and 5.1–6 min at 25:75) containing 0.1% (*v*/*v*) formic acid as a modifier at a flow rate of 0.3 mL/min. The sample injection volume was 10 μL and the column temperature was maintained at 40 °C. Quantitative analysis of baicalin was performed using Multi-reactions monitoring mode: +446.8→271.5 *m*/*z*.

### 2.14. HBV Infection and Mice Treatment

Each male BALB/c mouse (20 ± 1 g) was injected with 2 mL of PBS (containing 10 μg of the pHBV1.2 plasmid) via the tail vein [25]. On the next day, the mice infected with HBV were randomly assigned to 11 groups (i.e., model group; 5 mg/kg ETV; 1 mg/kg BA; 5 mg/kg BA; 10 mg/kg BA; 1 mg/kg BALP; 5 mg/kg BALP; 10 mg/kg BALP; 1 mg/kg BAA1; 5 mg/kg BAA1; and 10 mg/kg BAA1) and treated once a day. After continuous administration for 7 days, the mice were sacrificed, and their samples were collected for subsequent experiments. The serum was used for detecting the levels of HBsAg and HBV-DNA, and the liver tissues were subjected to HE staining.

### 2.15. Pharmacokinetics and Tissue Distribution Studies

The mice were randomly assigned to 3 groups (i.e., BA, BALP, and BAA1; 100 mg/kg BA; *n* = 3). BA was administered via gavage, and BALP or BAA1 were injected intraperitoneally. At 30, 60, 120, 240, 480, and 720 min after administration, the blood samples were collected in heparin microtubes and centrifuged at 3500 rpm for 10 min to obtain the plasma. The supernatant was used to analyze by HPLC the serum BA concentration-time profiles in the mouse. For the detection conditions of HPLC, please refer to Section 2.5., “The entrapment efficiency of BAA1”.

The treated mice were sacrificed to collect their livers, hearts, spleens, lungs, and kidneys for BA measurement 2 h post administration. The tissues were weighed and homogenized with saline at the ratio of 1:2 (weight: volume, *w*/*v*) and the spleen at 1:4 (weight: volume, *w*/*v*). Then, 200 μL of the sample was placed in a centrifuge tube and vortexed with 40 μL of concentrated hydrochloric acid (1 mol/L) for 1 min, followed by the addition of 20 μL of the internal standard (IS, 200 ng/mL carbamazepine methanol solution) and 800 μL of acetonitrile. The mixture was vortexed thoroughly, and the supernatant was obtained by centrifugation at 14,000 rpm for 10 min. The supernatant was dried in nitrogen at 36 °C, reconstituted in 100 μL of 70% methanol, vortexed thoroughly, and centrifuged at 15,000 rpm for 10 min to collect the supernatant for HPLC analysis [26]. The BA content in the final supernatant was analyzed by HPLC [15]. For the detection conditions of HPLC, please refer to Section 2.5.

### 2.16. Transcriptomics Analysis

pHBV1.2-transfected HepG2 cells were treated with BAA1 and BA, and the culture medium was refreshed with the sample solution every 2 days. After 96 h of incubation, the cell samples were collected for RNA-Seq. Briefly, the cell samples were prepared and submitted to Majorbio Bio-pharm Technology Corporation (Shanghai, China) for total RNA extraction, mRNA isolation and purification, sequencing library preparation, and sequencing. Next, total RNA was isolated and reversed into cDNA with the TRIzol reagent using the PrimeScript™ RT Reagent Kit with the gDNA Eraser (Perfect Real Time) (TaKaRa). Quality and quantity control was performed through fastp (Version 0.23.0) and faxtx-toolkit (Version 0.0.14). The libraries were sequenced by the Illumina platform (Majorbio Bio-pharm Technology Corporation, Shanghai, China). The quality control of the original sequencing data was monitored through SeqPrep to obtain high-quality quality control data (clean reads) so as to ensure the accuracy of the results of the subsequent analysis. The selected clean reads were aligned to the genomic reference sequence through the HISAT2 (Version 2.1.0), and the expression of the transcript was calculated using the RSEM (Version 1.3.3). Differential expression analysis of genes was performed using DESeq2. *p* < 0.05 was considered to indicate significant differential expression. All differential expression of the genes was analyzed by Gene Ontology (GO) enrichment and the Kyoto Genes and Genomes (KEGG) enrichment pathway. NCBI accession number: PRJNA 799795.

### 2.17. Statistical Analyses

GraphPad Prism for Windows (Version 8.0.2) was used to evaluate the statistical significance. Data were presented as a mean ± standard deviation of at least 3 independent experiments and analyzed using the two-tailed Student’s *t* test or one-way analysis of variance (ANOVA) followed by Bonferroni’s test. *p* < 0.05 was considered to indicate statistical significance.

## 3. Results

### 3.1. Characterization of BAA1

The freeze-dried liver-targeted BA liposomes (BAA1) were yellow and loose. Three batches of the prepared BAA1 were used for analyzing the particle size, PDI, zeta potential, and encapsulation efficiency. As shown in Figure 1, the average particle size of BAA1 was 104.5 ± 2.263 nm (Figure 1A), the PDI was 0.269 ± 0.011 (Appendix A), the zeta potential was −23.1 ± 0.651 mV (Figure 1B), and the encapsulation efficiency was 86.5 ± 5.2%. These results indicated that the novel ApoA1-modified liposomes were prepared successfully and were suitable for use in the experiments.

### 3.2. In Vitro Release and Cellular Uptake of BAA1

The dialysis method was used to evaluate the release abilities of free BA and the prepared BAA1 liposomes. As shown in Figure 2A, the release of BA from BAA1 was obviously lower than that from free BA. At 2, 4, and 8 h, the amounts of BA released from the liposomes were 40.6%, 67.3%, and 74.6%, respectively, while 78.6% BA was released from the free BA sample at 2 h. This in vitro release behavior of the BA liposomes was consistent with the previously reported results [15].

The cellular uptake experiment showed that a high amount of BA was taken up by the BAA1-treated HepG2 cells. As shown in Figure 2B, the cellular BA rapidly accumulated in the BAA1-treated cells, which was significantly higher than that in the BA-treated cells (4–12 h of incubation). These findings signify that the ApoA1-modified nanoparticles effectively promoted BA uptake by the HepG2 cells.

### 3.3. The Potent Inhibition of HBV Replication and Transcription by BAA1 in pHBV1.2 and rAAV-HBV-Transfected HepG2 Cells

The anti-HBV activities of BA and BAA1 were evaluated in the pHBV1.2-transfected HepG2 cells, which were transfected with pHBV1.2 plasmid to construct a transient HBV infection cell line. The cell viability assay showed that BA and BAA1 did not exhibit cytotoxicity when used at a concentration of 25–100 μM (Figure 3A) and were hence applied for anti-HBV evaluation. As shown in Figure 3B–F, compared with BA, BAA1 treatment resulted in much more potent inhibition of viral proteins (HBsAg and HBeAg), viral particles (HBV DNA), and HBV RNA transcription (total HBV-specific transcripts and pgRNA).

The anti-HBV efficacy of BAA1 was also evaluated in AAVDJ-transfected cells. The AAVDJ-vector carrying HBV B subtype is the major HBV genotype in China [27]. Potent inhibition of HBV transcription and replication were also observed in the BAA1-treated cells, which suggests that BAA1 could inhibit infection by different HBV genotypes (Figure 4).

These results imply that the ApoA1-modified nanoparticles could significantly improve the anti-HBV ability of BA.

### 3.4. Mediation of Enhanced Anti-HBV Activity of BA Liposomes by ApoA1 Modification

To clarify the role of ApoA1, the ordinary BA liposomes (BALP) and empty liposomes without BA (Empty LP) were further prepared to compare the effects of BA, BALP, and BAA1 on HBsAg, HBeAg, HBV DNA, and HBV RNA (total HBV-specific transcripts and pgRNA). As shown in Figure 5, compared with BA-containing liposomes, empty LP had no effect on HBsAg, HBeAg, and HBV DNA, suggesting that the liposome carrier and ApoA1 had no antiviral effects, and BA was the active ingredient. In addition, compared with BALP, BAA1 significantly inhibited HBV transcription and replication, suggesting that ApoA1 contributed to the anti-HBV activity of BAA1.

In addition, the role of ApoA1 was also verified by blocking the SR-B1 receptor with the SR-B1 antibody. As shown in Appendix A, blocking the receptor weakened the antiviral effect of BAA1 in pHBV1.2-transfected HepG2 cells, which indicates that ApoA1 played a crucial role in the anti-HBV efficacy of BAA1.

These results establish that the enhanced anti-HBV ability of BAA1 was not caused by the lipids and that ApoA1 modification improved the anti-HBV efficacy of the BA liposomes.

### 3.5. Augmentation of the ERα–HNF4α–HNF1α Signaling Axis by BAA1

HBV-cccDNA transcription was closely associated with the hepatocyte nuclear factors (HNFs). Our previous studies have confirmed that BA weakened the transcriptional activation of HNF4α on HNF1α and downregulated HNF1α expression, which inhibited HBV transcription and replication [7].

The possible influence of BAA1 on the transcription and expression of the related HNFs (HNF1α, FOXA2/HNF3β, and HNF4α) was further explored. As shown in Figure 6A,B, lower mRNA and protein amounts of HNF1α were observed in the BAA1-treated samples than in the BA-treated samples. However, there was no significant difference in HNF4 alteration of the BA- or BAA1-treated cells.

BA was observed to trigger the heterodimer formation of ERα and HNF4α to inhibit the transcriptional activation of HNF4α. As shown in Figure 6C–F, the Co-IP experiment confirmed that more ERα–HNF4α heterodimers were formed in the BAA1-treated cells than in the BA-treated cells. Significant ERα phosphorylation and transcription of ERα downstream genes (hB1F and PS2) were observed in the BAA1-treated cells.

The above results suggest that BAA1 might trigger the robust heterodimer of ERα and HNF4 to weaken transcription of HNF1α, thereby enhancing anti-HBV activity.

### 3.6. Transcriptomic Analysis

RNA-Seq was used to track the transcriptomic changes in the BA- and BAA1-treated cells. Principal component analysis (PCA) and Venn analysis showed that BAA1 yielded 456 genes that were differentiated from the model group (Figure 7A,B). The differentially expressed genes in each group were further analyzed using GO and KEGG enrichment. The results of the analyses revealed that BAA1 triggered the upregulation of the ER-related signaling pathways (vs. BA-treated samples), which is consistent with the above results (Figure 7C,E). The differentially expressed genes in the BA- and BAA1-treated cells were focused on the pathways related to sterol synthesis and energy metabolism, which might be affected by lipid composition (Figure 7D,F).

Above all, the ApoA1-modified liposomes augmented the ERα–HNF4–HNF1 axis to promote anti-HBV efficacy of BA.

### 3.7. Improvement of the Anti-HBV Efficacy of BA Liposomes In Vivo by ApoA1 Modification

To clarify the anti-HBV activity of BAA1 in vivo, an acute HBV infection mouse model was constructed by tail-vein, high-pressure injection of pHBV plasmids (pHBV1.2). The experimental procedure is depicted in Figure 8A. On Day 0, the mice received the high-pressure injection of the plasmids. On Day 1, blood was collected from the mice orbit for serum HBsAg antigen analysis. The mice were divided into 11 groups according to the HBsAg antigen levels, as follows: virus group; ETV (5 mg/kg); BA (1 mg/kg, 5 mg/kg, and 10 mg/kg); BALP (1 mg/kg, 5 mg/kg, and 10 mg/kg); and BAA1 (1 mg/kg, 5 mg/kg, and 10 mg/kg). BA was administered by gavage and BALP and BAA1 by intraperitoneal injection, once a day for 7 days. On Day 8, the mice were sacrificed to obtain blood and liver tissue samples for the experiments.

The results confirmed that BAA1 inhibited HBsAg and HBV DNA more significantly than the BA or BA liposome (BALP) treatments (Figure 8B,C). H&E staining further revealed that BA, BALP, and BAA1 treatments alleviated the acute liver tissue necrosis induced by HBV infection, which was comparable to the ETV treatment (Figure 8F).

The anti-HBV efficacy of BA was also explored in different routes of administration (oral and intraperitoneal). Serum HBsAg and HBV DNA were not inhibited effectively by 5 mg/kg BA (oral or intraperitoneal administration) but were significantly suppressed by the same dose of BAA1 (Figure 8D,E). These data suggest that ApoA1-mediated liver targeting might contribute to the enhanced anti-HBV efficacy of BA liposomes in vivo.

### 3.8. Pharmacokinetics and Tissue Distribution Study of BAA1 In Vivo

The plasma drug concentration and tissue distribution study of BA, BALP, and BAA1 (100 mg/kg) was performed in the mice. The pharmacokinetic results indicated that high absorption and long retention time of BA were observed in the BALP- and BAA1-treated samples (Figure 9A and Table 1). The results of tissue distribution experiments revealed that the highest amount of BA was accumulated in the liver tissues of the BAA1-treated mice, which signifies the possible liver-targeting effects of BAA1 (Figure 9B).

## 4. Discussion

BA has been reported to offer protection against several liver diseases [28] and a variety of viruses [29]. Recently, the anti-Zika activity [30] and anti-cor onavirus activity [31] of BA were also reported. However, poor bioavailability is the major problem that limits BA’s therapeutic applications. Liposome formation were successfully applied to improve the bioavailability of BA for treating various diseases. For example, BA nanoliposomes demonstrated better antitumor therapeutic effects in nude mice with human lung cancer in situ [32]. BA and BA liposomes were used in the treatment of nonalcoholic fatty liver and observed to exert good curative effects and reach high drug concentrations in the liver [26]. However, studies on BA liposomes have seldom focused on HBV therapy and liposome optimization. Therefore, we attempted to construct a liver-tissue-targeted modification of the liposomes to improve the bioavailability and antiviral efficacy of BA. In our previous work, recombinant ApoA1 protein was successfully overexpressed and applied for the delivery of doxorubicin hydrochloride to the liver, which indicates its potential for liver targeting [14]. Based on the characteristics of ApoA1, we prepared BAA1 and explored its characteristics and anti-HBV efficacy. The cellular uptake and tissue distribution experiments confirmed the enhanced bioavailability of the ApoA1-modified BA liposomes.

In HBV-infected cell and animal models, BAA1 showed more effective inhibition of HBsAg, HBeAg, and HBV DNA than BA treatment. Interestingly, BAA1 seems to be more potent in inhibiting HBV RNA levels in genotype B when compared to genotype C, which may be due to the different replication abilities and transcriptional activities of the different genotypes of HBV. Studies have reported that the genome replication ability and transcriptional activity of 3.5 kb RNA of HBV genotype B were significantly stronger than those of genotype C [33]. Furthermore, we found that in the pHBV1.2 [34]-transfected HepG2 cells, BAA1 treatment inhibited HBsAg and HBeAg more significantly than HBV RNAs. The reasons for this may be multifaceted. HBsAg was translated by 2.4/2.1 kb RNA, while HBeAg was translated by 3.5 kb RNA. As a result, it is possible that the viral RNA levels were not consistent with protein levels. In addition, because the detected viral proteins were extracellular secreted antigens by the ELISA method, their levels may be inconsistent with the intracellular RNA levels by the RT-PCR method. This also suggests that BAA1 might play a role in inhibiting antigen secretion, which needs further investigation. In addition, we believe that there may be some minor HBV plasmids in the cell supernatant, but most of the viral DNA should be present in the viral particles, and the residual HBV plasmids do not contribute on the same order of magnitude as viral DNA. We will find out more accurate quantitative methods to analyze HBV RNA in rcDNA in the future.

Additional experiments were designed to clarify how BAA1 exerted the improved anti-HBV ability. Empty liposomes without BA and BA liposomes (BALP) without ApoA1 were prepared to further explore the influence of lipid composition on HBV transcription and replication. The results indicated that the anti-HBV efficacy was not due to the lipid components but was closely related to BA. The anti-HBV efficacy of BAA1 was higher than that of free BA and BALP, which meant that the liver-targeted delivery benefited HBV therapy. The possible influence of different BA preparations on the HNF4α–HNF1α signaling pathway was also explored. In our previous study, we established that BA could regulate the transcription and translation of HBV via the HNF4α–HNF1α-dependent signaling pathway, which might be related to the phosphorylation of ERα [35]. In this study, promotion of ERα phosphorylation, transcription of ERα downstream genes, and Erα–HNF4α heterodimer formation were observed in the BAA1- and BA-treated cells. BAA1 demonstrated significant interfering effects on the ERα–HNF4–HNF1 axis when compared with the BA treatment. Furthermore, transcriptomics analysis was used to clarify the possible influence of different BA preparations. The data showed that BAA1 mainly triggered the transcription of ER-related genes, which is consistent with the findings of the RT-PCR and Western blot experiments in this work. In BA- and BAA1-treated samples, the differentially expressed genes were mainly related to sterol synthesis and energy metabolism pathways, which might be due to the membrane components of BAA1. These findings imply that liposome encapsulation does not alter the antiviral signaling pathway of BA in an ERα–HNF4–HNF1-dependent manner.

In summary, liver-targeted BA liposomes (BAA1) were successfully prepared. Moreover, it was proven that the enhanced bioavailability could significantly promote the anti-HBV efficacy of BA in vitro and in vivo, thereby opening promising avenues for the application of BA in anti-HBV treatment.

## Figures and Tables

**Figure 1 biomedicines-10-00900-f001:**
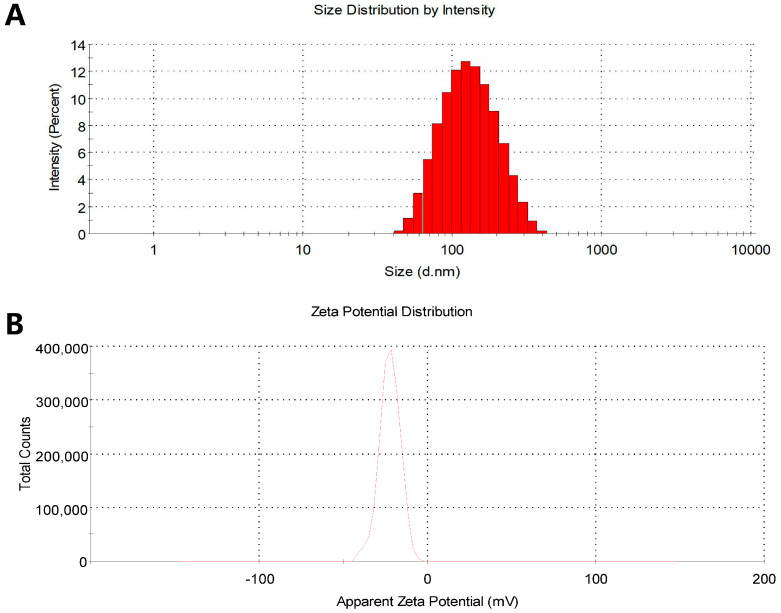
Characteristics of the liver-targeted baicalin liposomes (BAA1). (**A**) The particle size was measured by using the Malvern Zetasizer Nano ZS90 (Malvern Instruments Ltd., Malvern, UK). (**B**) Zeta potential was measured by using the Malvern Zetasizer Nano ZS90 (Malvern Instruments Ltd., Malvern, UK).

**Figure 2 biomedicines-10-00900-f002:**
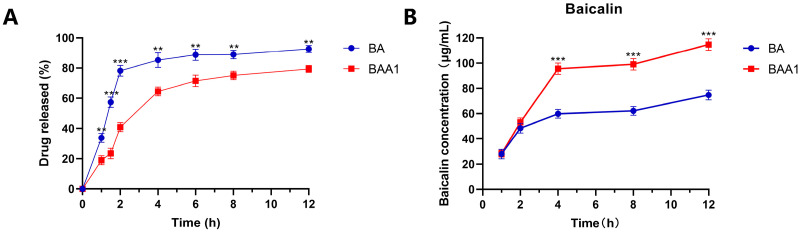
Cumulative percentage drug release and cellular uptake from BA and BAA1. (**A**) Cumulative percentage drug release from free BA and BAA1. (**B**) The cellular uptake of BA and BAA1 in HepG2 cells. Data are mean ± SD, *n* = 3; ** *p* < 0.01, *** *p* < 0.001.

**Figure 3 biomedicines-10-00900-f003:**
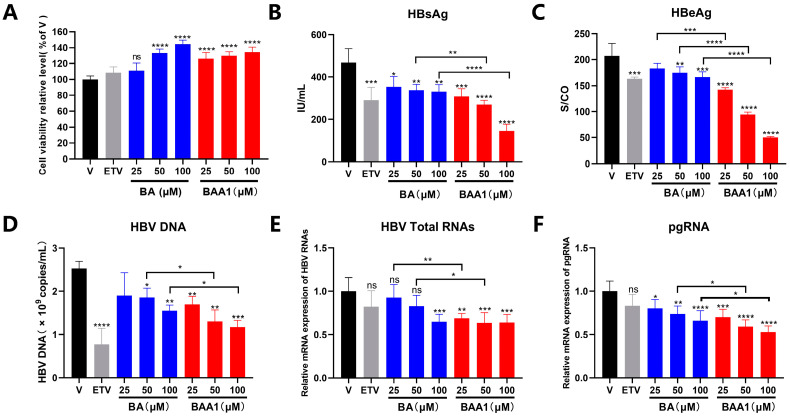
Liver-targeted baicalin liposomes (BAA1) and baicalin (BA)-inhibited HBV replication in HepG2 cells. (**A**) pHBV1.2-transfected HepG2 cells were treated with ETV (10 μM), BA (25–100 μM), and BAA1 (25–100 μM) for 2 days. Cell viability was detected using the CCK-8 kit. (**B**,**C**) HBsAg and HBeAg in the cell supernatants were detected by using ELISA assay kits. (**D**) HBV-DNA in the culture supernatants was detected by qPCR. (**E**,**F**) HBV total RNAs and pgRNA were determined by qRT-PCR. Data are presented as mean ± SD, *n* = 3; * *p* < 0.05, ** *p* < 0.01, *** *p* < 0.001, and **** *p* < 0.0001 vs. control. ns, not significant.

**Figure 4 biomedicines-10-00900-f004:**
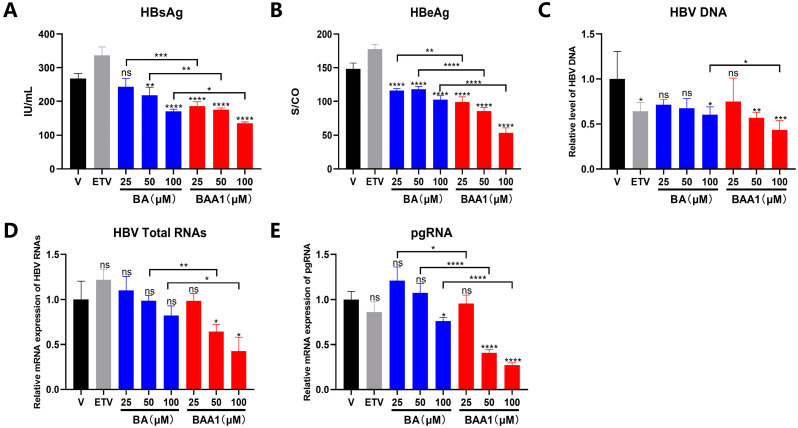
Liver-targeted baicalin liposome (BAA1) and baicalin (BA)-inhibited HBV replication in AAVDJ-transfected HepG2 cells. (**A**,**B**) AAVDJ-transfected HepG2 cells were treated with ETV (10 μM), BA (25–100 μM), and BAA1 (25–100 μM). HBsAg and HBeAg in the culture supernatants were detected by using ELISA kits. (**C**) HBV-DNA (HBV virion) in the culture supernatants was detected by qPCR. (**D**,**E**) HBV total RNAs and pgRNA were determined by qRT-PCR. Data are presented as mean ± SD, *n* = 3; * *p* < 0.05, ** *p* < 0.01, *** *p* < 0.001, and **** *p* < 0.0001 vs. control. ns, not significant.

**Figure 5 biomedicines-10-00900-f005:**
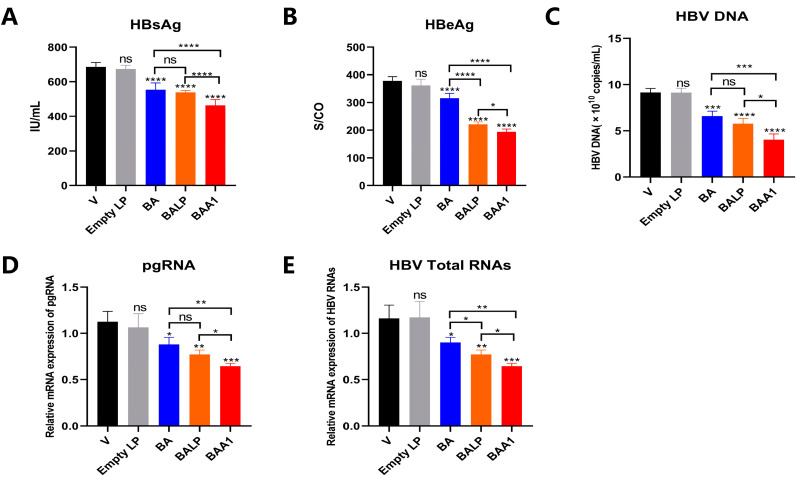
ApoA1 modification enhanced the anti-HBV effects of BA liposomes. pHBV1.2-transfected HepG2 cells were treated with empty liposomes (LP), BA liposomes (BALP, 50 μM), BA (50 μM), and BAA1 (50 μM) for 2 days. (**A**,**B**) HBsAg and HBeAg in the culture supernatants were detected by using ELISA kits. (**C**) HBV-DNA (HBV virion) in the culture supernatants was detected by qPCR. (**D**,**E**) HBV total RNAs and pgRNA were determined by qRT-PCR. Data are presented as the mean ± SD, *n* = 3; * *p* < 0.05, ** *p* < 0.01, *** *p* < 0.001, and **** *p* < 0.0001 vs. control. ns, not significant.

**Figure 6 biomedicines-10-00900-f006:**
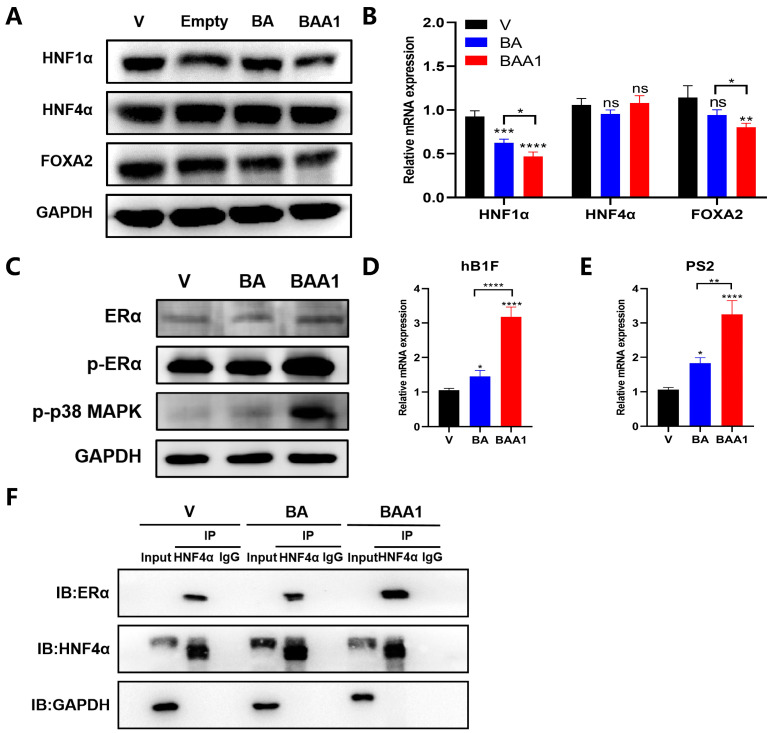
Liver-enriched transcription factors were influenced by BAA1 treatment. pHBV1.2-HepG2 cells were treated with BA (50 μM) or BAA1 (50 μM) for 4 days. (**A**,**C**) HNF1α, HNF4α, FOXA2, Erα, p-ERα, and p-p38 MAPK expression in pHBV1.2-transfected HepG2 cells (Western blotting). (**B**,**D**,**E**) the relative mRNA levels of HNF1α, HNF4α, FOXA2, hB1F, and PS2 (qRT-PCR). Data are presented as mean ± SD, *n* = 3; * *p* < 0.05, ** *p* < 0.01, *** *p* < 0.001, and **** *p* < 0.0001 vs. control. ns, not significant. (**F**) The combination of ERα and HNF4α in pHBV1.2-transfected HepG2 cells was detected by Western blotting.

**Figure 7 biomedicines-10-00900-f007:**
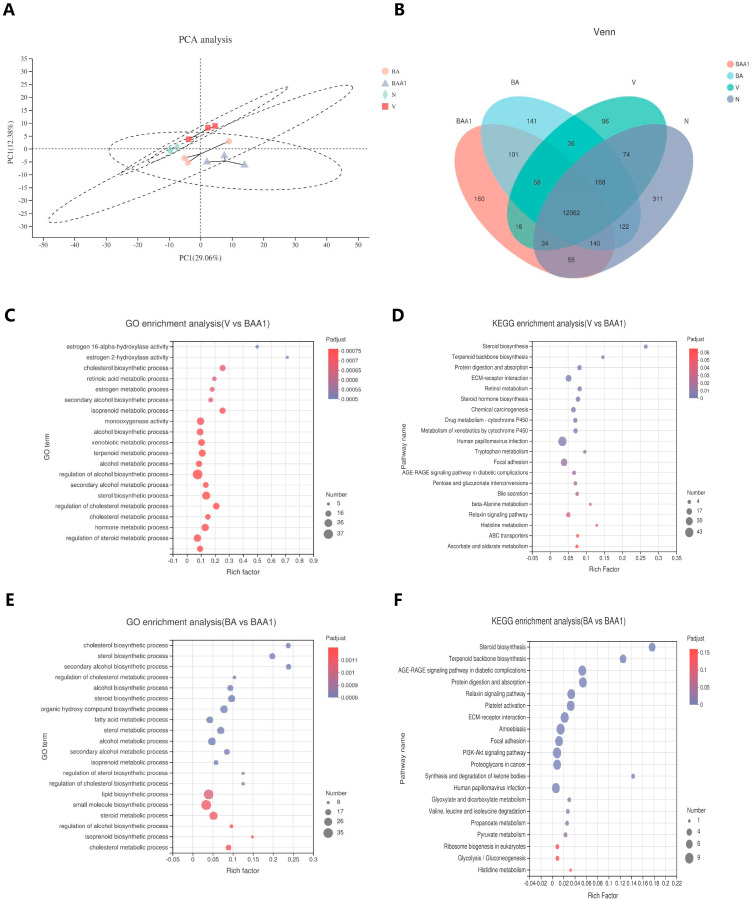
Transcriptomics analysis of BA- and BAA1-treated cells. pHBV1.2-transfected HepG2 cells were treated with 50 μM of BA or BAA1 for 4 days and subjected to transcriptomics analysis. (**A**) PCA analysis, (**B**) Venn analysis, and (**C**,**D**) GO and KEGG enrichment analyses between V and BAA1. (**E**,**F**) GO and KEGG enrichment analyses between BA and BAA1. The transcriptomics data were analyzed on the free online platform (www.majorbio.com, accessed on 24 January 2022).

**Figure 8 biomedicines-10-00900-f008:**
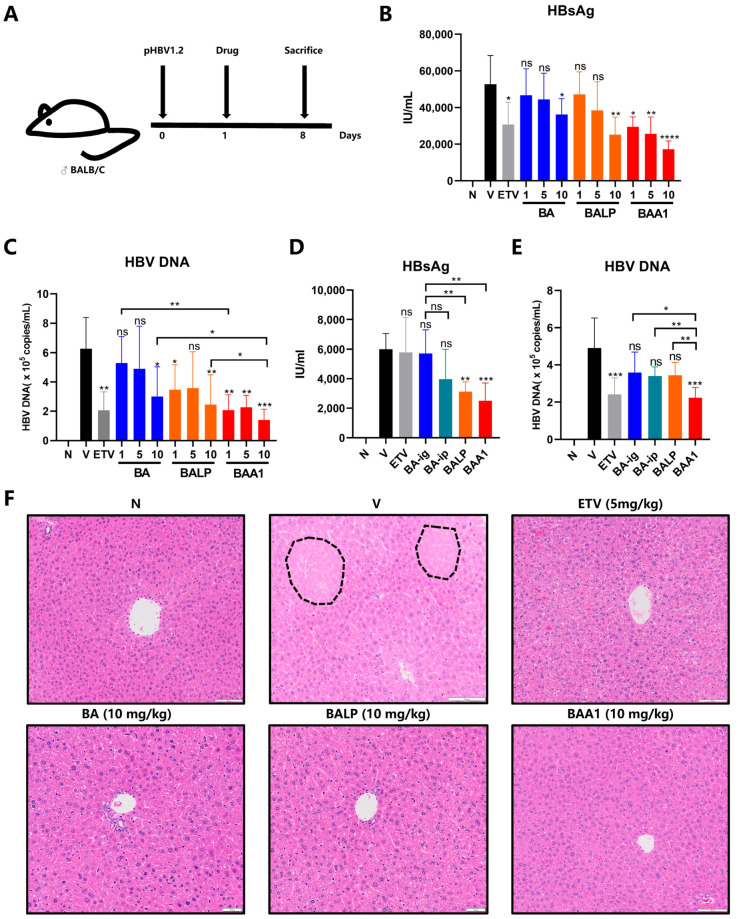
Anti-HBV efficacy of different baicalin preparations in vivo. (**A**–**C**) HBV transient mice were hydrodynamically injected with pHBV1.2 plasmids. The infected mice received oral administration of BA (1, 5, 10 mg/kg); ETV (5 mg/kg); intraperitoneal injection of BALP (1, 5, 10 mg/kg); or BAA1 (1, 5, 10 mg/kg) once a day for 7 days. The mice were sacrificed for their serum HBsAg and HBV DNA, with detection by ELISA and qPCR, respectively. Data are presented as mean ± SD, *n* = 3; * *p* < 0.05, ** *p* < 0.01, *** *p* < 0.001, and **** *p* < 0.0001 vs. control. ns, not significant. (**D**,**E**) The infected mice received oral administration of BA (5 mg/kg); ETV (5 mg/kg); intraperitoneal injection of BA (5 mg/kg); BALP (5 mg/kg); or BAA1 (5 mg/kg) once a day for 1 week. The mice were sacrificed for their serum HBsAg and HBV DNA. with measurements by ELISA and qPCR, respectively. Data are presented as mean ± SD, *n* = 3; * *p* < 0.05, ** *p* < 0.01, *** *p* < 0.001, and **** *p* < 0.0001 vs. control. ns, not significant. (**F**) HE staining of the liver tissues (200× magnification). Area of necrosis (black dotted line). Scale bar = 100 μm.

**Figure 9 biomedicines-10-00900-f009:**
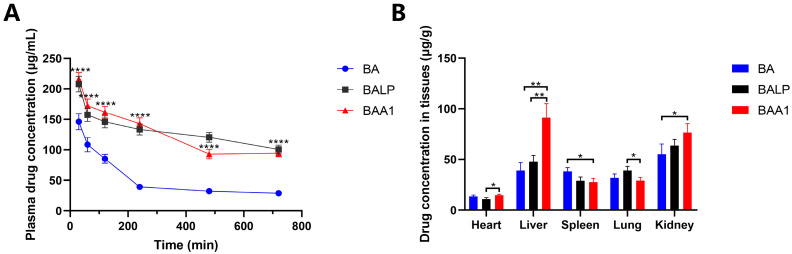
Pharmacokinetics and tissue distribution of BA, BALP, and BAA1 at the dose of 100 mg/kg. (**A**) The plasma BA concentration-time profiles in the mouse plasma were detected by HPLC. Data are presented as mean ± SD, *n* = 3; **** *p* < 0.0001. (**B**) Drug distribution in each tissue 2 h after administration, detected by HPLC. Data are presented as mean ± SD, *n* = 3; * *p* < 0.05, ** *p* < 0.01.

**Table 1 biomedicines-10-00900-t001:** Main pharmacokinetic parameters of plasma in the normal mice.

Parameters	Units	BA	BALP	BAA1
AUC_0-t_	mg/L*min	32,446.896 ± 2833.222	87,550.161 ± 4794.225	84,249.757 ± 3453.486
AUC_0-∞_	mg/L*min	45,990.555 ± 13,235.327	246,086.047 ± 17,232.716	157,887.132 ± 33,642.481
MRT_0-t_	min	283.610 ± 2.9314	347.976 ± 2.102	325.323 ± 2.277
MRT_0-∞_	min	689.113 ± 211.102	1606.579 ± 85.404	999.485 ± 103.403
t_1/2z_	min	411.193 ± 242.653	1093.433 ± 52.416	601.197 ± 152.636
V_z/F_	L/kg	1.196 ± 0.360	0.643 ± 0.033	0.543 ± 0.032
C_Lz/F_	L/min/kg	0.002	0.001	0.001
C_max_	mg/L	108.451 ± 9.493	157.420 ± 8.945	173.132 ± 8.026

BA, BALP, and BAA1 were each administered both orally and by intraperitoneal injection at 100 mg/kg, and the data were expressed as the mean ± SD (*n* = 3). Abbreviations: AUC, area under the plasma drug concentration-time curve; MRT, mean residence time; t_1/2z_, statistical moments half-life; V_z/F_, volume of distribution; C_Lz/F_, clearance rate; C_max_, peak concentration.

## Data Availability

Transcriptomics data were submitted to the NCBI’s Sequence Read Archive (SRA). Accession number NCBI: PRJNA 799795.

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
