# Peer review of "Liver-Targeted Nanoparticles Facilitate the Bioavailability and Anti-HBV Efficacy of Baicalin In Vitro and In Vivo"

_biomedicines, 2022, doi:10.3390/biomedicines10040900_

Round 1

Reviewer 1 Report

In this manuscript titled “Liver-target nanoparticle facilitates the bioavailability and anti-HBV efficacy of baicalin in-vitro and in-vivo”, the authors tried to improve the bioavailability of baicalin via usage of modified nanoparticles. The authors evaluated the pharmacological properties of the apolipoprotein A1 (ApoA1)-modified liposomes (BAA1) in vitro and in vivo, and demonstrated that BAA1 has a superior anti-HBV effect compared to baicalin in vitro and in vivo. Besides, the authors also demonstrated the potential mechanism of how BAA1 achieved the superior anti-HBV effect via western blot and RNA-seq. Overall, this is a very interesting and timely study, which could potentially have a significant impact in the HBV field. However, some key control is missing in some experiments, and this manuscript is poorly written so that key information is missing. Therefore, a major revision is needed.

Major issues:

  1. The results from RNA-seq is not convincing (Figure 7A). First of all, the resolution of figure 7A is too low, a high-resolution picture is needed. Even more importantly, it looks like the transcriptome from all the groups, including N, V, BA, BAA1, is pretty similar and PC1+PC2 only explains less than 50% of variety. It makes me wonder whether the quality of RNA-seq is good enough, whether the authors can generate a robust HBV infection in the sequenced cells and whether there is true transcription difference among different treatments. The authors need to include a positive control in the RNA-seq to make sure they can detect transcriptome changes, such as a ETV treated group as the authors used in other experiments.

  1. For readers outside of HBV field, the full name of ETV should appear somewhere in this manuscript.

  1. The results between figure 3 and figure 4 are not very consistent. For example, ETV has a clear anti-HBV effect in figure 3, but didn’t have much effect in figure 4. The authors should explain the reasons generating such inconsistency in discussion or somewhere else in this manuscript. More importantly, if ETV doesn’t have much effect in figure 4, why did the authors use it as a positive control in figure 4? The authors should add another true positive control in the assays performed in figure 4.

  1. Why is the empty LP control not included in figure 5C, D, E?

  1. The introduction and method are not well written, so that key information is missing and some results look confusing. For example, in figure 3A, why do the BA and BAA1 treated cells have a significantly higher cell viability than 100%? Is it because BA and BAA1 can stimulate cell proliferation? Is there any known off-target effect of BA? How was the cell viability assay performed? The method section didn’t describe it at all. The introduction and method need to be re-written to include more detailed information regarding BA, BAA1 and the assays performed in this manuscript.

Reviewer 2 Report

The study developed an ApoA1 modified liposome for targeted delivery of baicalin in liver cells. The authors showed the effect of this modification on anti-HBV activity both in vitro and in vivo. The study is novel, and the conclusions are relevant to the questions asked. However, some concerns need to be addressed.

Major points

  • Figure 3D: HBV DNA is detected in culture supernatants by qPCR. Are the authors referring to cccDNA here? The culture supernatant will have transfected HBV DNA contamination. How are the authors getting rid of the transfected DNA? Please elaborate in the methods section.
  • In their previous study, the authors show that BA can inhibit HBV transcription. In this study, modification of BA to BAA1 results in a significant reduction in HBsAg and HBeAg (protein levels) at 100uM, but the RNA levels don’t seem to vary much (Figure 3Band 3C vs 3E and 3F). Can the authors comment on this observation?
  • BAA1 seems to be more potent in inhibiting HBV RNA levels in genotype B when compared to genotype C. Can the authors comment on the genotype specificity of BAA1?
  • Figure 5: BAA1 seems only modestly (10-20% more than BALP) effective in inhibiting HBV than BALP. Also, does the empty LP control used here contain ApoA1? If not, please include an empty LP-ApoA1 control. Why is the empty LP bar missing in figure 5C-F?
  • Line 292-293; BAA1 treated samples have lower amounts of HNF1a. Is this observation true for cells not infected with HBV? In other words, is this observation independent of HBV infection?
  • Figure 6A: Western blot band intensity for HNF1a between the “Empty” lane and “BAA1” lane is comparable. So, if the empty LP does not contain ApoA1, the reduction in H1NFa cannot be attributed to ApoA1. Please provide western blot quantification for differences observed.
  • Line 301: BAA1 enhances anti-HBV activity and lowers HNF1a protein levels. These two can be independent events. To claim that the two events are related the authors should perform additional experiments. For eg. – Does overexpression or knockdown of HNF1a change HBV protein and RNA levels?

Minor points

  • Methods section: Please provide additional details on the amount of plasmid transfected in HepG2 cells.
  • Line 116: Please correct this line. RNA was isolated using TriZol reagent and reverse transcribed to cDNA using….
  • Figure 2B- Chart header- Please correct the spelling.
  • Line 239-240- resulted in much more potent inhibition of viral protein… compared to what?

Round 2

Reviewer 1 Report

All concerns resolved

Author Response

Dear Reviewer,

Many thanks for your recognition. And thank you very much for your valuable comments and suggestions on our work to make our manuscripts more professional.

Thank you and best regards.

Sincerely,

Xunlong Shi

Reviewer 2 Report

Figure 3D: HBV DNA is detected in culture supernatants by qPCR. Are the authors referring to cccDNA here? The culture supernatant will have transfected HBV DNA contamination. How are the authors getting rid of the transfected DNA? Please elaborate in the methods section. √Thank you for your critical comments and the helpful suggestions.

HBV cccDNA is present in the nucleus, and HBV DNA detected in culture supernatants by qPCR refers to capsid-packaged viral DNA produced by reverse transcription of pgRNA. To get rid of the possible residual plasmid contamination, we removed as much residual plasmid as possible by washing with PBS and then changing the cell culture medium after plasmid transfection for 6h. The related description has been added to the methods section.

I am afraid that washing with PBS will not get rid of transfected DNA contamination. As the authors are measuring encapsidated relaxed circular DNA, one way to approach this problem is to DNase treat the supernatant and deactivate the DNase using a DNase deactivating resin followed by lysing the virus and quantitating the HBV rc-DNA using qPCR.

OR

One can also capture the virus in the culture supernatant in an HBsAg ELISA plate after DNase treatment of the sup. After capture, the plate can be washed several times and DNA can be extracted for quantification. The detailed protocol can be found here (https://www.sciencedirect.com/science/article/pii/S0255085721002619)
